# Mechanism of Lithotripsy by Superpulse Thulium Fiber Laser and Its Clinical Efficiency

**Vladimir Lekarev [1], Alim Dymov [1], Andrey Vinarov [1], Nikolay Sorokin [2], Vladimir Minaev [3], Nikita Minaev [4,*], Svetlana Tsypina [4] and Vladimir Yusupov [4]**

[1] Institute for Urology and Reproductive Health, Sechenov University, stroenie 1, 2, Bolshaya Pirogovskaya St., 119992 Moscow, Russia; lekarev_bat@mail.ru (V.L.); alimdv@mail.ru (A.D.); avinarov@mail.ru (A.V.)

[2] Lomonosov Moscow State University, University Clinic, stroenie 53, 1, Leninskie gory, 119991 Moscow, Russia; nisorokin@mail.ru

[3] "IRE-Polus" Ltd., bld. 3, 1, pl. Akad. Vvedensky, Fryazino, 141190 Moscow Region, Russia; minaev46@mail.ru

[4] Institute of Photon Technologies, Federal Scientific Research Centre «Crystallography and Photonics», Russian Academy of Sciences, 2, Pionerskaya St., 108840 Troitsk, Moscow, Russia; tsypina@yandex.ru (S.T.); iouss@yandex.ru (V.Y.)

\* Correspondence: minaevn@gmail.com



**Featured Application: The data presented in the article show the high efficiency of the superpulse thulium fiber laser for laser lithotripsy. We clarified the mechanism of action for the fragmentation of a stone phantom.**

**Abstract:** Thulium fiber laser with a wavelength of 1.94 μm is widely used in urology for lithotripsy. This paper studies the mechanism of lithotripsy and evaluates its clinical efficiency using the superpulse thulium fiber laser with a wavelength of 1.94 μm and a peak power of 500 W. An experimental setup was developed to study the mechanism of lithotripsy. The superpulse thulium fiber laser (TFL) with a wavelength of 1.94 μm with a peak power of 500 W (FiberLase U2 from "IRE-POLUS" Ltd., Fryazino, Moscow Region, Russia) was used for the lithotripsy of stone phantoms (BegoStone). The processes were recorded with a high-speed camera. The acoustic signals registered during lithotripsy were studied with wideband and needle hydrophones. The main mechanism of lithotripsy performed by using superpulse TFL was thermal cavitation in the water-filled pore space and thermal destruction of the phantom. During the clinical application of the superpulse thulium fiber laser, the high efficiency of laser lithotripsy was established. The performed optical and acoustic studies showed that the mechanism of the destruction of stones was based on the synergic effect of the explosive boiling of water in the pore space of the stone, and its thermal destruction is associated with the heating of the stone to several hundred degrees with laser radiation.

**Keywords:** superpulse thulium fiber laser; laser lithotripsy mechanism; thermal cavitation; thermal destruction

## 1. Introduction

During the past decades, surgical treatment for urinary stone disease changed a lot because of significant technological developments and the improvement of medical equipment [1]. The appearance of laser technologies brought innovative approaches to the surgical treatment of urinary stones. Presently, laser treatment is a generally accepted method of lithotripsy [2,3]. The efficiency of lithotripsy of different types of stones with minimum damage to the surrounding soft tissues depends on the parameters of laser radiation (wavelength, pulse duration, and power) and the properties of

stones (optical, mechanical, and chemical). The prevailing mechanism of disintegration/lithotrypsy (photothermal or photoacoustic/photomechanical) depends on the intensity, energy, and duration of a pulse. It is believed that laser radiation with a pulse duration $\tau > 10$ µs causes a significant increase in the temperature in the area of laser exposure in cases with minimal acoustic waves. In such cases, the material of the stone is removed by means of evaporation, melting, thermomechanical tension, and/or chemical decomposition (photothermal mechanism). On the contrary, a laser with a short pulse duration (<10 µs) produces a shock wave that mechanically fragments the stone (photoacoustic mechanism) [4].

According to current EAU (the European Association of Urology) guidelines, holmium laser lithotripsy (wavelength $\lambda = 2.1$ µm) is the gold standard [5]. In the 2000s, researchers' attention was drawn to the application of thulium (Tm)-doped fiber lasers (thulium fiber laser) with a wavelength of 1.94 µm for lithotripsy [6,7]. It was established that holmium laser ablation rates were lower than for the thulium fiber laser (TFL). The TFL also produced a greater percentage by the mass of stone dust (fragments less than 0.5 mm), which can lead to lower complication rates. According to Hardy L.A. et al. 2019, 7% stones treated with a holmium laser are completely fragmented in ≤5 min compared 60% stones treated with TFL [8]. TFL provides greater versatility and control of pulse parameters than the holmium: yttrium aluminum garnet (Ho:YAG) laser, including advanced pulse-shaping capabilities. A comparison of ablation volumes indicated that the TFL induced significantly higher (twofold) ablation than the Ho:YAG laser [9]. Similar data were obtained by [10]. It was reported that TFL with an optimal wavelength and long pulse duration resulted in faster stone ablation and lower retropulsion in comparison to the Ho:YAG laser. It was also shown that no substantial difference in the maximum temperature rise of water around stones was observed between the two laser systems. However, fiber burnback was more pronounced for the Ho laser, especially at higher power settings [10]. These studies demonstrate that the TFL is a promising alternative laser for lithotripsy, producing higher stone ablation rates, smaller stone fragments, and respectively lower complication rates than the holmium laser.

It is believed that the main mechanism of lithotripsy with 2 µm range laser radiation was photothermal. It was suggested that the mechanism of laser lithotripsy included two processes. First, a quick heating of water outside the stone leads to the formation of a closed cavity with compressed vapor that expands, collapses, and creates a shock wave that destroys the stone. Second, water contained in micropores and microfissures (around 10%) expands under quick heating from laser exposure, which leads to high pressure because of its incompressibility. Additional pressure occurs when water boils, evaporates, and condenses, including the cases when blast waves are formed. All these processes lead to the disintegration of the stone [11–13].

The effectiveness of laser exposure significantly depends on the wavelength because it affects the water absorption coefficient. A higher water absorption coefficient leads to faster heating of water and the formation of a vapor bubble. For the radiation used in a urology Ho:YAG laser with $\lambda = 2.1$ µm, the coefficient of water absorption is near 30 cm$^{-1}$, which significantly differs from the local peak of absorption in this spectral range (Figure 1). For TFL radiation with wavelength $\lambda = 1.94$ µm, the absorption coefficient of water radiation is significantly higher and equals $129.8 \pm 0.5$ cm$^{-1}$ [3,14].

To improve the results of laser lithotripsy and reduce its duration, we used a superpulse TFL with $\lambda = 1.94$ µm and a peak power of up to 500 W (FiberLase U2) [10,15]. The performed preliminary comparative experimental studies on the efficiency of the proposed Ho:YAG laser in lithotripsy showed a high efficiency of TFL lithotripsy in all fragmentation regimes [16].

The main mechanisms of stone disintegration with this new thulium fiber laser are of great practical and scientific interest. The investigation of the processes that are observed during laser action is necessary for the further optimization of laser lithotripsy.

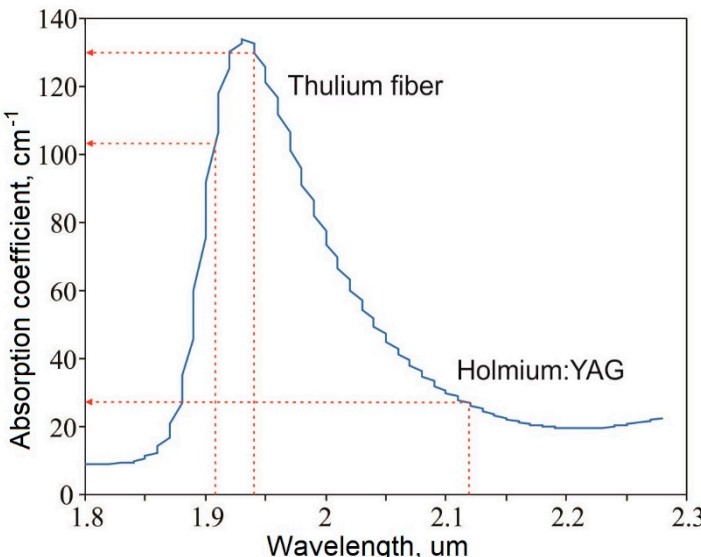

**Figure 1.** Dependence of the absorption coefficient of water (22 °C) on the wavelength of radiation in the area of the local peak ≈2 μm [14].

Experimental studies that include a high-speed optical camera and acoustic measurements in the wide range of frequency are optimal in terms of the investigation of processes that occur during laser action, in particular, stone disintegration [17].

The aim of the present study was to investigate the mechanism of lithotripsy and to evaluate its clinical efficiency using a superpulse TFL with a wavelength of 1.94 μm and a peak power of 500 W.

## 2. Materials and Methods

Experimental studies were performed in a transparent cuvette filled with 0.9% saline solution (NaCl) using the superpulse thulium fiber laser. We used end-firing laser fibers with a core diameter of 600 μm. The lithotripsy experiments were carried out on cubic stone phantoms (BegoStone) with an edge size of 5 mm. The stone phantoms were exposed to laser radiation with the following parameters: average power $P_{av}$ = 10 W, pulse duration $\tau$ = 0.4 ms, pulse energy $E_p$ = 0.2 J, and peak power $P_p$ = 500 W, which provided the so-called dusting mode of stone disintegration. In total, three series of experiments were carried out, in which 7 BegoStones were used.

We installed a wideband hydrophone 8103 (Bruel&Kjaer, Nærum, Denmark) with a 0.1 Hz–180 kHz band (sensitivity −211 dB re 1 V/μPa) in the cuvette ≈3 cm from the fiber end perpendicularly to the fiber optical axis. For the registration of high-frequency acoustic signals, a needle hydrophone (Precision Acoustics, Dorchester, UK) with a diameter of 1 mm and with a 10 kHz–50 MHz bandwidth (sensitivity −241 dB re 1 V/μPa) equipped with a pre-amplifier was installed ≈1 cm from the end of the fiber. Acoustic signals obtained with hydrophones were recorded by a four-channel digital storage oscilloscope GDS 72,304 (GW Instek, Taipei, Taiwan) with a 300 MHz bandwidth.

An optical registration of laser-induced processes near the fiber end in water was performed with a high-speed camera Fastcam SA-3 (Photron, Tokyo, Japan) (4000 fps, frontal light through the cuvette, shadowgraph technique).

The fragment of the installation with a transparent cuvette and the layout of the configuration of its separate elements during the stone phantom laser exposure are presented in Figure 2. The stone phantom was fixed on a special stationary holder. The phantom was exposed to the laser in a contact mode when the fiber end firmly meets its surface.

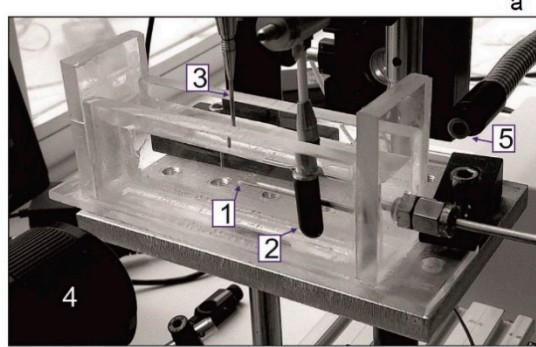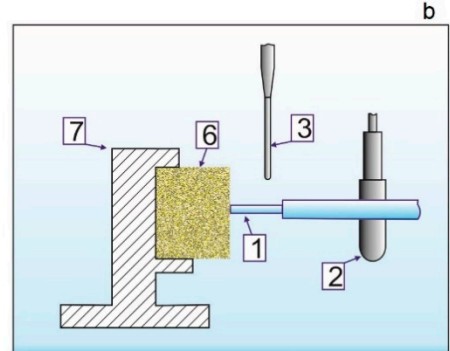

**Figure 2.** Fragment of experimental laser-induced hydrodynamic processes near the fiber end (**a**) and the layout of the configuration of the elements under the phantom laser exposure (**b**). 1—fiber end (tip), 2—wide-band hydrophone 8103, 3—needle hydrophone, 4—high-speed camera lens Fastcam SA-3, 5—frontal light, 6—phantom of the stone, 7—holder.

The evaluation of the clinical efficiency of laser radiation with $\lambda = 1.94$ μm and 500 W maximum peak power was based on the results of its clinical application [15,18]. In the Institute for Urology and Reproductive Health of the Sechenov University, retrograde flexible nephrolithotripsy, mini percutaneous nephrolithotripsy, endoscopic ureterolithotripsy, and cystolithotripsy are performed using the follow equipment: fibers with core diameters of 200, 400, and 600 μm, respectively. Retrograde flexible nephrolithotripsy is performed using a disposable digital flexible ureterorenoscope with an external diameter of the tube 9.5 Fr, an operating channel of 3.6 Fr, and an angle of deflection of 270° in both directions (upwards and downwards). During mini percutaneous nephrolithotripsy, a rigid nephroscope 16.5/17.5 Ch with an offset eyepiece 120 is used. Endoscopic ureteral lithotripsy is performed with a rigid uretheroscope Richard Wolf 8/9.8 Ch with an offset eyepiece 120 and Karl Storz 7/12 Ch. Contact cystolithotripsy is performed with an operating element KUNTZ with a resectoscope tube 26 Ch.

Retrograde intrarenal surgery is performed using two modes of laser radiation, wherein the pulse energy ranged from 0.15 to 0.8 J and average power ranged from 15 to 30 W. The dusting mode with a high repetition rate (30–200 Hz) is optimal to provide good stone disintegration with the formation of very fine particles and create conditions for their free passage. Mini percutaneous nephrolithotripsy is highly efficient in the laser settings with pulse energy that ranged from 0.15 to 4 J and average power that ranged from 15 to 40 W. For ureteral stones, the pulse energy varied from 0.15 to 0.8 J and the average power varied from 6 to 30 W. Taking into account a limited manipulation space in the ureter, we use a relatively low repetition rate (10–30 Hz) for the improvement of the visibility during the procedure. The fragmentation of stones in the bladder is performed with high-energy pulses (0.5–6 J) and an average power from 15 to 40 W. The postoperative stone-free rate (residual stone assessment) can be evaluated with CT images 1 week after surgery. Postoperative ureteral stenosis is evaluated with clinical data and cystoscopy view 2–3 months after surgery.

## 3. Results and Discussion

Both acoustic and optical experiments were aimed at obtaining high-quality information about the ongoing processes. They showed that a typical optical picture of stone destruction is observed under laser action in the superpulse mode. Fragments of such a typical picture during high-speed shooting are shown in Figure 3. The signals recorded by hydrophones did not differ qualitatively in different experiments, either. Fragments of typical acoustic signals are shown in Figure 4. Analysis of the obtained optical and acoustic patterns made it possible to propose hypotheses explaining the high efficiency of the superpulse thulium fiber laser.

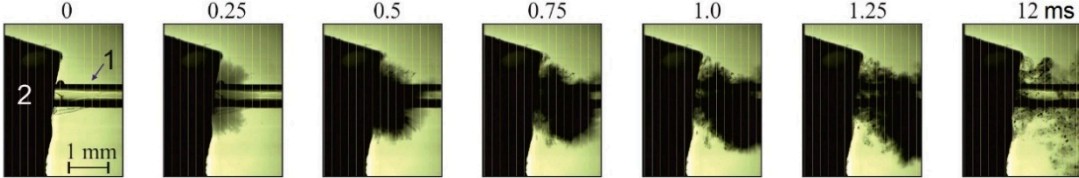

**Figure 3.** Typical pictures from a high-speed camera taken during the fragmentation of the stone phantom in a dusting regime. 1—laser fiber, 2—phantom of the stone. The time scale is in ms.

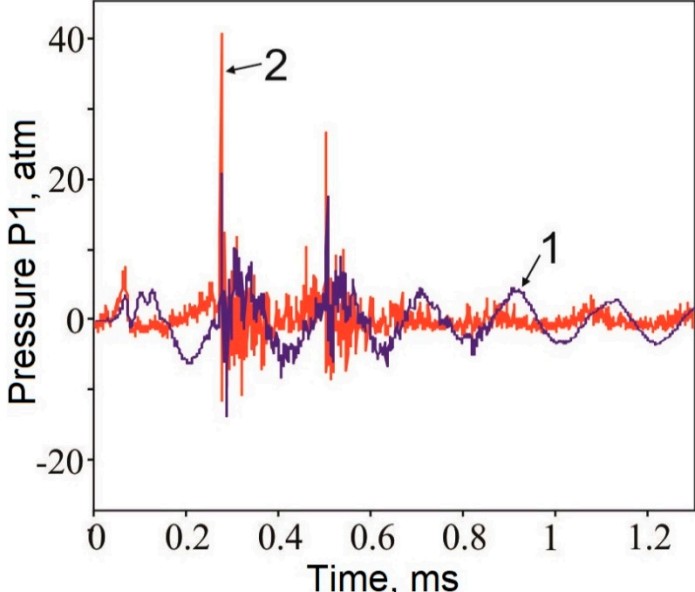

**Figure 4.** Fragments of typical acoustic signals registered by a wide-band hydrophone (1) and needle hydrophone (2) registered during the fragmentation of the stone phantom in the dusting regime. The pressure was recalculated for the distance of 100 μm.

### 3.1. Optical and Acoustic Tests

Figure 3 shows pictures taken by a high-speed camera during the disintegration of the stone phantom in the dusting mode. During laser pulse exposure, an outburst of a cloud of microscopic fragments of the material into the liquid is observed. Initially, the speed of this outburst reaches a minimum of ≈3.2 m/s, and 1 ms later, it decreases to 2.5 m/s. The frame 12 ms after the beginning of the phantom ablation clearly shows separate particles of the material outburst from the area of exposure. The size of the majority of dust-like fragments varies within the range of 1–10 μm. The largest single particles reached up to 50 μm in diameter.

Figure 4 shows fragments of the acoustic signal registered by wideband and needle hydrophones during the laser exposure of the stone in the dusting regime. The curves contained several strong pulses with a duration of not more than 5 μs. The acoustic noise in the wavetrain started with a wider and significantly lower amplitude of prepulse registered by both hydrophones. After a spike of acoustic energy, fading oscillations are observed. The curve of a wideband hydrophone (1 in Figure 4) clearly shows nearly harmonic fading oscillations with a frequency of 5 kHz.

The amplitudes of maximal pulses (recalculated for the distance of 100 μm) registered by a needle hydrophone were ≈40 atm and ≈27 atm. It should be noted that there was a 0.4 ms delay between the last pulse and the prepulse, which corresponded to the duration of the laser pulse.

The tests showed that during the disintegration of the stone phantom in the dusting regime, the following occurred: (1) there was an active burst-like outburst of microscopic (1–50 μm) fragments of the material into the liquid (pictures from a high-speed camera in Figure 3); (2) the outburst was

associated with a generation of strong short pulses of acoustic signals with maximal amplitudes of dozens of atmospheres (results of the acoustic survey, Figure 4).

Possible physical mechanisms explaining the registered results are described below.

The main characteristic of the medium, which determines the degree of the impact of laser radiation, is the absorption coefficient. For numerous water-containing tissues, this coefficient is defined by the radiation absorption in water. For $\lambda = 1.94$ μm, it is $129.8 \pm 0.5$ cm$^{-1}$ [14]. When it comes to stones, it was shown that for the nearest infrared range of wavelengths, the absorption coefficient hardly depended on the stone types [19]. The dependence of the absorption coefficient in water on the radiation wavelength in the area of the local peak in the range of $\approx 2$ μm is presented in Figure 1. It can be seen that the TFL radiation is in the area of peak values, while the radiation of the Ho:YAG laser is in the edge of the peak, where the absorption coefficient is significantly lower (for 2.12 μm, $\mu_a = 26.1 \pm 0.2$ cm$^{-1}$).

There are several mechanisms observed during the laser lithotripsy. Photomechanical ablation occurs in the regime of short pulses (<1 μm) and high intensity of laser radiation (>$10^8$ W/cm$^2$). This mechanism of destruction prevails in cases with a so-called "stress confinement": $\tau < \delta/c$, where $\tau$—the duration of the laser pulse, $\delta = 1/\mu_{eff}$—the depth of optical penetration, $\mu_{eff}$—the coefficient of effective attenuation that includes the absorption and the impact of radiation scattering, $c$—the sound velocity in tissue. Physically, this means that during the pulse duration, the tension provoked by fast heating cannot leave the area where the main part of the laser energy is absorbed. In such conditions, laser energy can effectively transform into mechanical or acoustic energy because of a thermoelastic response [19].

Photothermal ablation is typical for lasers with relatively long pulses (>1 μm) and a relatively low intensity of laser radiation (<<$10^8$ W/cm$^2$). The criterion of this process efficiency is described by an inequality: $\tau < \delta^2/4\alpha$, where $\alpha$—the coefficient of thermal conductivity of the medium. Physically, this means that during the pulse duration, the heat produced in the medium as a result of laser energy absorption cannot leave the area where the main part of the laser energy is absorbed. In cases when long pulses $\tau >> \delta/c$ are applied, the pressure relief from the heated area occurs during laser-radiation heating, and the pressure peak $\Delta P$ is $\tau \cdot c\alpha$ times lower than during the realization of the conditions for photomechanical ablation [20].

In cases with even longer pulses and continuous laser radiation, the mechanism of tissue destruction associated with thermal cavitation is observed. It is characterized by the transfer of water to the near-critical or super-critical area of temperatures and pressures. Initially, due to the absorption of laser radiation, an overheated area is gradually formed in water. When the temperature comes closer to the spinodal (critical temperature $T_C \approx 374$ °C for the atmospheric pressure [21]), fluctuations lead to the explosive boiling of the heated liquid with the formation of a quickly expanding steam–gas cavity (bubble) [17,22]. When the cavity reaches its maximal size, it quickly collapses and causes a pressure increase and cumulative jet and shock wave formation [23,24], which leads to the destruction of the tissue [25].

In cases with an enclosed volume of pore space, the transition to the point of thermal cavitation is observed along with a simultaneous increase in the pressure [26]. When the water reaches the area of the spinodal ($T_C \approx 374$ °C), explosive water boiling occurs, and the water in pores transforms into vapor compressed to critical pressure ($P_C \approx 22$ MPa) [26,27]. When critical parameters for the tissue are exceeded, such pressure peaks lead to the explosive destruction and formation of shock waves. Shock waves also contribute to the destruction of adjoining tissues and the outburst of particles of the fragmented material from the area of ablation [28]. The performed tests showed that for the majority of stones, the pore sizes varied from 1 to 100 μm [29].

It should be noted that the above-described process of the destruction associated with the explosive boiling of water in pores of the material can be realized due to the Moses effect [30] without tight contact between the end of laser fiber and the tissue. Due to this effect, a vapor–gas bubble is formed between the fiber end and the surface of the tissue, freely transmitting laser radiation.

In the present study, we used a laser with $\lambda = 1.94$ µm, $\tau = 0.4$ ms, $E_p = 0.2$ J, $P_p = 500$ W (dusting regime). It is easy to calculate that in cases when the fiber diameter is 600 µm, the intensity of laser radiation in the pulse at the end of the fiber will be $1.8 \cdot 10^5$ W/cm$^2$, in other words, $<<10^8$ W/cm$^2$, which is necessary for the realization of the mechanism of photomechanical ablation. The depth of laser radiation penetration in the stone material is $\delta = 1/\mu_{eff} \approx 0.1$ mm. This distance is covered within $\approx 50$ ns by a sound wave that spreads at a velocity of $\approx 1500$ m/s. Thus, another condition for the realization of the mechanism of photomechanical ablation by the laser pulse duration ($\tau < \delta/c$) is not fulfilled, too.

When it comes to photothermal ablation, the main criterion for its realization, shorter pulse duration ($\tau = 0.4$ ms) in comparison with $\delta^2/4\alpha \approx 10^{-8}$ (and the coefficient of thermal conductivity of water $\alpha = 0.143$ m$^2$/s), was not fulfilled, too.

Taking into account the above-described facts, we suggested that in the present study, the destruction of the stone phantom under laser radiation was performed primarily due to thermal cavitation in the water-filled pore space of the phantom [21,31,32]. From the point of view of further physical processes, it is important to note that thermal cavitation occurs in a closed microscopic space [33]. The jump of pressure that is observed in this volume during fast water heating (1.6 MPa at $T= 200$ °C and 4 MPa at 250 °C) can lead to the disintegration of the phantom. Such laser-induced shock processes during the explosive boiling of water near the fiber end were studied using optoacoustic methods [33,34].

It is important to mention that the effect of pulse pressure, which is observed during the explosive boiling of water in the micropores of the phantom, occurs in the conditions of heated phantom. The degree of the destruction of the material $\Omega(t)$ under laser exposure is well-described by the Arrhenius integral [35]:

$$\Omega(t) = \varsigma \int_0^{Time} \exp[-E_a/RT(t)]dt \tag{1}$$

where $\zeta$ (s$^{-1}$)—the frequency factor, *Time*—the full time of heating, $E_a$ (J/mol)—the energy of activation; $R$ (8.32 J/K mol)—the universal gas constant, and $T(t)$ (K)—the absolute temperature of the material. The degree of destruction $\Omega(t)$ exponentially depends on temperature and linearly on the time of heating. It should be noted that at $\Omega(t) = 1$, 63% of the material gets destroyed.

During the time of laser pulse exposure, the temperature in the water-filled stone tissue (in a contact regime when the fiber end firmly touches its surface) increases by the value calculated by the formula:

$$\Delta T = \frac{C\mu_a F}{\rho c_p} \exp(-\mu_a z) \tag{2}$$

where C—the pore water content, $\mu_a$—the coefficient of absorption (cm$^{-1}$), $\rho$—density (kg/m$^3$), $c_p$—heat capacity at constant pressure (J/kgK), $F = P_p \cdot \tau$—fluence or density of energy (J/cm$^2$), and z—depth from the fiber end into the material. Depending on certain physicochemical characteristics of the stone, its heating can lead to the weakening of its structure and destruction. Note that the stone is heated exclusively through the water in the pore space (the stone itself does not absorb laser radiation).

If the temperature does not exceed critical values ($T_{crit}$) during heating, the speed of the material destruction $\Delta\Omega/\Delta t = \zeta \exp(-E_a/RT_{crit})$ is insignificantly low. On the contrary, at $T > T_{crit}$, the speed of destruction increases exponentially. The critical temperature depends on the certain composition of the stone and its structure. For magnesium ammonium phosphate, it is 100 °C, for calcium oxalate monohydrate, it is 206 °C, for cystine, it is 264 °C, and for uric acid, it is 360 °C [36]. In other words, during laser heating of the material within the temperature of the spinodal for water (critical temperature $T_c = 374$ °C), the temperature for the majority of stones will exceed $T_{crit}$.

Thus, it can be concluded that the thermal destruction associated with the laser heating of the stone during lithotripsy, along with the explosive boiling of water in pores, is an important mechanism of its destruction.

*3.2. Clinical Case*

A clinical case of retrograde intrarenal lithotripsy with a TFL is presented below.

Patient B (female, 53 years old) was hospitalized to the Institute for Urology and Reproductive Health of the Sechenov University in November 2019 with complaints of a left back pain.

Computed tomography (CT) scans showed a stone in the pelvis of the left kidney 1.9 × 1.7 cm (density 1380 HU) (Figures 5 and 6).

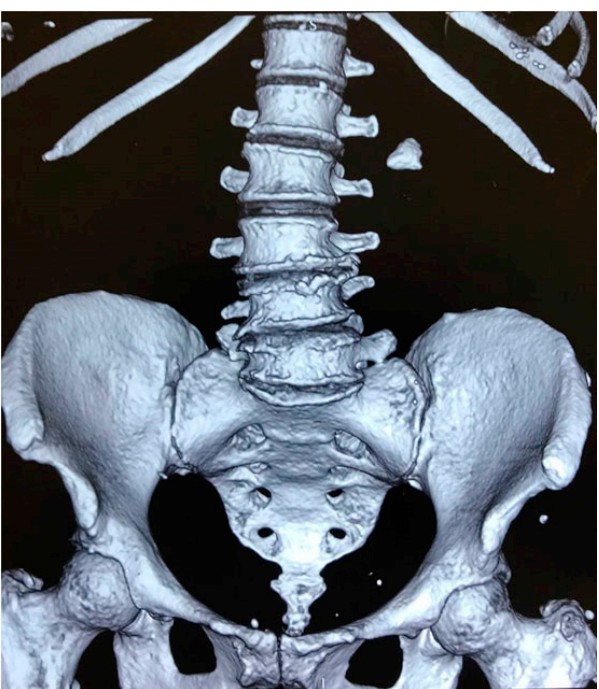

**Figure 5.** CT scan. The stone is in the projection of the left kidney, 1.9 × 1.7 cm, density 1380 HU.

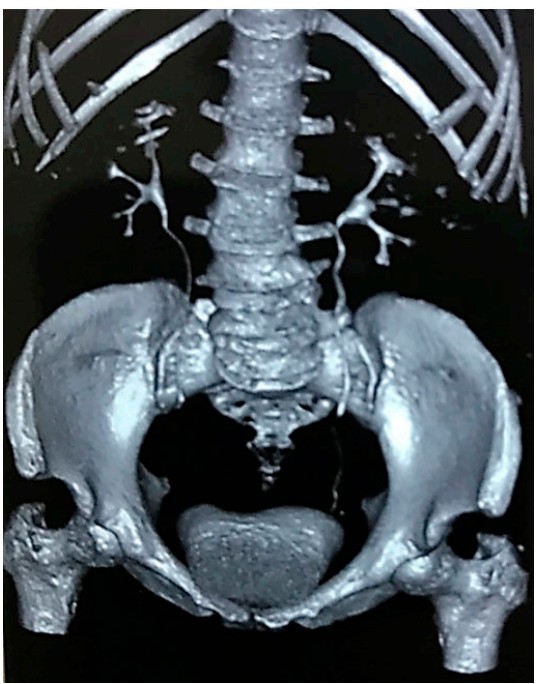

**Figure 6.** CT urogram.

Taking into account the size and density of the stone, surgical treatment was indicated.

In November 2019, the patient underwent cystoscopy. The tube ureteral access 10/12 Ch was placed using a hydrophilic guidewire in the left upper urinary tract (UUT). The tube was used to insert a disposable flexible ureterorenoscope. A 2 cm single yellow stone with a rough surface was visualized in the pelvis. Lithotripsy was performed with a TFL with a wavelength of 1.94 μm and maximal peak power of 500 W in the dusting mode (0.15 J, 200 Hz) using a 200 μm laser fiber to the fine dust. The control examination of the calices–pelvis system did not reveal stone fragments >0.2 mm. The left UUTs were drained with a ureteral JJ stent Ch7.

There were no complications observed during the postoperative period. The stent was removed 2 weeks after the surgery. Control CT was performed 3 months after the surgery and did not reveal stones in the left UUT (Figure 7).

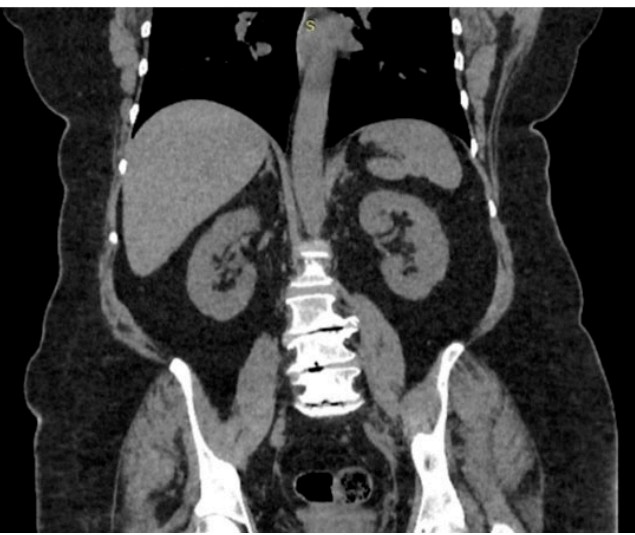

**Figure 7.** CT scan. No stones in the left upper urinary tract (UUT).

## 4. Conclusions

The performed optical and acoustic tests showed that during the disintegration of the stone phantom in the dusting mode, an active explosion-like outburst of microscopic fragments of the material was observed associated with the generation of powerful short acoustic signals with maximal amplitudes (dozens of atmospheres). It was established that the mechanism of destruction was based on the synergic effect of explosive boiling of water in the pore space of the material and thermal destruction of the stone associated with laser heating to several hundred degrees. The high clinical efficiency of the new thulium fiber laser with a wavelength of 1.94 μm with maximal peak power up to 500 W was shown.

**Author Contributions:** Conceptualization, V.M. and A.V.; methodology, V.L. and N.S.; validation, V.L. and A.D.; formal analysis, V.Y. and V.M.; investigation, V.Y. and N.S.; resources, N.M.; data curation, S.T. and V.Y.; writing—original draft preparation, V.M., V.Y. and V.L.; writing—review and editing, N.M., S.T.; visualization, V.Y.; supervision, A.V. and N.S.; project administration, V.M. All authors have read and agreed to the published version of the manuscript.

**Funding:** The study was supported by the Ministry of Science and Higher Education within the framework of the state project of the Federal Scientific-Research Center "Crystallography and Photonics" of the RAS on the development of new approaches to bioablation and Russian Foundation for Basic Research (Project No.18-29-06056) in part of laser structuring of materials.

**Conflicts of Interest:** The authors declare no conflict of interest.

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
