# Peer review of "Mechanism of Lithotripsy by Superpulse Thulium Fiber Laser and Its Clinical Efficiency"

_applsci, doi:10.3390/app10217480_

Round 1

Reviewer 1 Report

The authors have submitted an interesting manuscript on an experimental study on the mechanisms of stone disintegration by the new thulium fibre laser.

Some issues, however, should be addressed:

The clinical sections (Materials and Methods and Results are redundant as it is an experimental study. It should not be mixed with clinical cases as these cases to not give information on the mechanism of stone disintegration.

The authors did not report how many experiments have been done. How many series with how many BegoStones have been carried out? The experiments should be evaluated statistically b appropriate tests.

Author Response

Dear Reviewer,

We are grateful for your careful reading of our article.

  1. The clinical sections (Materials and Methods and Results) are redundant as it is an experimental study. It should not be mixed with clinical cases as these cases to not give information on the mechanism of stone disintegration.

Answer:

Thanks for your comment. You are right that the article is mainly devoted to clarifying the mechanism of action of the new laser. However, since the superpulse thulium fiber laser is a recent development, we believe that demonstrating its clinical efficacy in this article is appropriate. Following your recommendation, this part has been corrected.

  1. The authors did not report how many experiments have been done. How many series with how many BegoStones have been carried out? The experiments should be evaluated statistically b appropriate tests.

Answer:

Thank you for your comment. We partially agree with the remark. A total of three series of experiments were carried out in which 7 BegoStones were used. Both acoustic and optical experiments were aimed at obtaining high-quality information about the ongoing processes. They showed that a typical optical picture of stone destruction is observed under laser action in the superpulse mode. Fragments of such a typical picture during high-speed shooting are shown in Fig. 3. The signals recorded by hydrophones did not differ qualitatively in different experiments either. Fragments of typical acoustic signals are shown in Fig. 4. Analysis of the obtained optical and acoustic patterns made it possible to propose hypotheses explaining the high efficiency of the superpulse thulium fiber laser.

Changes:

- Legends to Figures 3 and 4 indicate that typical images are presented.
- A sentence has been introduced into the text: "In total, three series of experiments were carried out, in which 7 BegoStones were used."

Good luck, and most importantly, good health!

On behalf of authors, Dr. Nikita Minaev

Reviewer 2 Report

1)General comments

The authors estimated the mechanism of lithotripsy and to evaluate its clinical efficiency using the superpulse thulium fiber laser with a wavelength of 1.94 µm and a peak power of 500 W. They concluded the mechanism of the destruction of stones was based on the synergic effect of the explosive boiling of water in the pore space of the stone and its thermal destruction associated with the heating of the stone to several hundred degrees with laser radiation.

The reviewer generally agrees with the conclusion.

However, there are several issues need to improve. The reviewer would like suggests several issues as follows;

2)Specific comments for revision

a)Major

#1 Please describe the differences between superpulse thulium fiber laser and holumium laser in the discussion part. Has stone ablation speed been reported with the Holumium laser?

Is there a difference between superpulse thulium fiber laser and holumium laser in complications?

Are there any reports of differences in lithotripsy efficiency or temperature rise of water around stones between superpulse thulium fiber laser and holumium laser?

#2 There are no details on the results of the surgery by superpulse thulium fiber laser (p3, line112-116).

What is the stone free rate and operation time? How is complication rate such as ureteral injury and postoperative ureteral stenosis?

b)Minor

#1 p8 line275, “On November 14, 2019, the patient underwent cystoscopy.” The date of surgery is not good for patient identification; therefore, it's better to list only month rather than the date.

Author Response

Dear Reviewer,

We are grateful for your careful reading of our article and appreciation of our article.

  1. a) Major

#1 Please describe the differences between superpulse thulium fiber laser and holumium laser in the discussion part. Has stone ablation speed been reported with the Holumium laser?

Is there a difference between superpulse thulium fiber laser and holumium laser in complications?

Are there any reports of differences in lithotripsy efficiency or temperature rise of water around stones between superpulse thulium fiber laser and holumium laser?

Answer: Thanks for your comment. We agree with all the comments, added data comparing two lasers in clinical practice.

Changes: It was established that holmium laser ablation rates were lower than for TFL. The TFL also produced a greater percentage by mass of stone dust (fragments less than 0,5 mm), what can leads to less complication rates. According to Hardy L.A. et al. 2019 7% stones treated with Holmium laser can completely fragmented in ≤ 5 minutes compared 60% stones treated with TFL. TFL provides greater versatility and control of pulse parameters than the Ho:YAG laser, including advanced pulse-shaping capabilities. Comparison of ablation volumes indicated that the TFL induced significantly higher (twofold) ablation than the Ho:YAG laser. The similar data was obtained by Andreeva V. et al. 2019. It was reported that TFL with optimal wavelength and long pulse duration resulted in faster stone ablation and lower retropulsion in comparison to the Ho:YAG laser. It was also shown that no substantial difference in the maximum temperature rise of water around stones was observed between the two laser systems. However, fiber burnback was more pronounced for the Ho laser, especially at higher power settings. These studies demonstrate that the TFL is a promising alternative laser for lithotripsy, producing higher stone ablation rates, smaller stone fragments and respectively less complication rates than the Holmium laser.

#2 There are no details on the results of the surgery by superpulse thulium fiber laser (p3, line112-116).

What is the stone free rate and operation time? How is complication rate such as ureteral injury and postoperative ureteral stenosis?

Answer: Thanks for your comment. The purpose of this section in the materials and methods was to describe the equipment that is used in lithotripsy. Necessary corrections for understanding have been made. The description of the results of this study in the materials and methods is not intended. We attach articles here that describe the results of interest. Also, links to them were added to the text of the article.

1) Dmitry Enikeev, Mark Taratkin, Roman Klimov, Yuriy Alyaev, Leonid Rapoport, Magomed Gazimiev, Dmitry Korolev, Stanislav Ali, Gagik Akopyan, Dmitry Tsarichenko, Irina Markovina  Eugenio Ventimiglia, Evgenia Goryacheva, Zhamshid Okhunov, Francis A Jefferson, Petr Glybochko, Olivier Traxer. Thulium-fiber laser for lithotripsy: first clinical experience in percutaneous nephrolithotomy. World J Urol. 2020 Feb 27. doi: 10.1007/s00345-020-03134-x.

2) Dmitry Enikeev, Mark Taratkin, Roman Klimov, Jasur Inoyatov, Camilla Azilgareeva, Stanislav Ali, Dmitry Korolev, Mariela Corrales, Olivier Traxer, Petr Glybochko. Superpulsed Thulium Fiber Laser for Stone Dusting: In Search of a Perfect Ablation Regimen-A Prospective Single-Center Study. J Endourol. 2020 Jul 15. doi: 10.1089/end.2020.0519.

Changes:

The evaluation of the clinical efficiency of laser radiation with ?=1.94 µm and 500 W maximum peak power was based on the results of its clinical application. In Institute for Urology and Reproductive Health of the Sechenov University retrograde flexible nephrolithotripsy, mini percutaneous nephrolithotripsy, endoscopic ureterolithotripsy, and cystolithotripsy are performed using follow equipment: fibers with core diameters of 200 µm, 400 µm, and 600 µm, respectively. Retrograde flexible nephrolithotripsy is performed using a disposable digital flexible ureterorenoscope with an external diameter of the tube 9.5 Fr and operating channel 3.6 Fr, angle of  deflection 270° in both directions (upwards and downwards). During mini percutaneous nephrolithotripsy, a rigid nephroscope 16.5/17.5 Ch with an offset eyepiece 120 is used. Endoscopic ureteral lithotripsy is performed with a rigid uretheroscope Richard Wolf 8/9.8 Ch with an offset eyepiece 120 and Karl Storz 7/12 Ch. Contact cystolithotripsy is performed with an operating element KUNTZ with a resectoscope tube 26 Ch.

Retrograde intrarenal surgery is performed using two modes of laser radiation, wherein the pulse energy ranged from 0.15 to 0.8 J and average power ranged from 15 to 30 W. The dusting mode with high repetition rate (30-200 Hz) is optimal to provide good stone disintegration with formation of very fine particles and create conditions for their free passage. Mini percutaneous nephrolithotripsy is highly efficient in the laser settings with pulse energy that ranged from 0.15 to 4 J and average power from 15 to 40 W. For ureteral stones pulse energy varied from 0.15 to 0.8 J and average power from 6 to 30 W. Taking into account a limited manipulation space in the ureter, we use relatively low repetition rate (10–30 Hz) for the improvement of the visibility during the procedure. The fragmentation of stones in the bladder is performed with high energy pulses (0.5–6 J) and average power from 15 to 40 W. Postoperative stone-free rate (residual stone assessment) can be evaluated with CT images 1 week after surgery. Postoperative ureteral stenosis is evaluated with clinical data and cystoscopy view 2-3 months after surgery.

b)Minor

#1 p8 line275, “On November 14, 2019, the patient underwent cystoscopy.” The date of surgery is not good for patient identification; therefore, it's better to list only month rather than the date.

Answer: Thanks for the correction. We agree with the comments, the necessary changes have been made.

Changes: In November 2019, the patient underwent cystoscopy

Good luck, and most importantly, good health!

On behalf of authors, Dr. Nikita Minaev

Reviewer 3 Report

No comments

Author Response

Dear Reviewer,
We are grateful for your appreciation of our article.
Good luck, and most importantly, good health!
On behalf of authors, Dr. Nikita Minaev

Reviewer 4 Report

The paper describes the results of research in vitro and clinical conditions on the new thulium fiber laser. The methodology of working in the in vitro part is correct.

The clinical part is very poor and is limited to one case, which does not allow drawing further conclusions, but only suggests that the laser works well in clinical conditions.The literature is adequate, despite the citation of a large number of papers authored by the authors of this report. This is done in a reasonable manner.The manuscript may be linguistically flawed.

Author Response

Dear Reviewer,

We are grateful for your careful reading of our article.

The clinical part is very poor and is limited to one case, which does not allow drawing further conclusions, but only suggests that the laser works well in clinical conditions.

Answer:  Thank you for your comment. The aim of the study was to describe the physical aspects of lithotripsy with a clinical example of the use of a thulium laser. Analysis of clinical data is beyond the scope of the task and will be carried out as part of another study.

The manuscript may be linguistically flawed.

Answer: Thank you for your comment. We turned to our colleague, who corrected the text of the manuscript. If necessary, we are ready to send the manuscript for correction to English editing from MDPI.

Good luck, and most importantly, good health!

On behalf of authors, Dr. Nikita Minaev

Round 2

Reviewer 1 Report

the authors have picked up the majority of the recommendations. Concerning the numbers of experiments I recommend to write  in the manuscript he same text as in the answer letter:

"A total of three series of experiments were carried out in which 7 BegoStones were used. Both acoustic and optical experiments were aimed at obtaining high-quality information about the ongoing processes. They showed that a typical optical picture of stone destruction is observed under laser action in the superpulse mode. Fragments of such a typical picture during high-speed shooting are shown in Fig. 3. The signals recorded by hydrophones did not differ qualitatively in different experiments either. Fragments of typical acoustic signals are shown in Fig. 4. Analysis of the obtained optical and acoustic patterns made it possible to propose hypotheses explaining the high efficiency of the superpulse thulium fiber laser."

Author Response

Dear Reviewer,

We are grateful for your careful reading of our article.

Thanks for the correction. We agree with the comments, the necessary changes have been made. Additional text has been added in to the manuscript.

Good luck, and most importantly, strong health!

On behalf of authors, Dr. Nikita Minaev

Reviewer 2 Report

Everything the reviewer pointed out was properly improved.